# Light-Responsive Soft Robot Integrating Actuation and Function Based on Laser Cutting

**DOI:** 10.3390/mi15040534

**Published:** 2024-04-16

**Authors:** Ben Jia, Changbo Liu, Yi Zhang, Yujin Tan, Xuecheng Tian, Yuanyuan Cui, Yuan Deng

**Affiliations:** 1School of Materials Science and Engineering, Beihang University, Beijing 100191, China; sy2101125@buaa.edu.cn (B.J.); tanyujin@buaa.edu.cn (Y.T.); 2Laboratory of Intelligent Sensing Materials and Chip Integration Technology of Zhejiang Province, Hangzhou Innovation Institute of Beihang University, Hangzhou 310051, China; sy2143234@buaa.edu.cn (Y.Z.); zy2143328@buaa.edu.com (X.T.); yuanyuancui@buaa.edu.cn (Y.C.); 3Research Institute for Frontier Science, Beihang University, Beijing 100191, China

**Keywords:** stimuli-responsive, soft robot, laser cutting, programmable deformation, light actuation

## Abstract

Soft robots with good deformability and adaptability have important prospects in the bionics and intelligence field. However, current research into soft robots is primarily limited to the study of actuators and ignores the integrated use of functional devices and actuators. To enrich the functions of soft robots and expand their application fields, it is necessary to integrate various functional electronic devices into soft robots to perform diverse functions during dynamic deformation. Therefore, this paper discusses methods and strategies to manufacture optical stimuli-responsive soft actuators and integrate them into functional devices for soft robots. Specifically, laser cutting allows us to fabricate an optically responsive actuator structure, e.g., the curling direction can be controlled by adjusting the direction of the cutting line. Actuators with different bending curvatures, including nonbending, can be obtained by adjusting the cutting depth, cutting width, and the spacing of the cutting line, which makes it easy to obtain a folded structure. Thus, various actuators with complex shape patterns can be obtained. In addition, we demonstrate a fabrication scheme for a worm-like soft robot integrated with functional devices (LEDs are used in this paper). The local nonbending design provides an asymmetric structure that provides driving power and avoids damage to the functional circuit caused by the large deformation during movement. The integration of drive and function provides a new path for the application of soft robots in the intelligence and bionics field.

## 1. Introduction

Natural organisms have the unique ability of self-feedback and are stimulus-driven, e.g., the crawling of worms and heliotropism of sunflowers, which makes them well adapted to complex and changing external environments. This strong environmental adaptability has inspired researchers to develop biomimetic robots with similar functions by imitating the movements of various creatures. In recent years, great progress has been made in the development of soft robots based on flexible driving materials [1,2,3,4]. Differing from traditional rigid robots, soft robots have high degrees of freedom and adaptability during deformation and use. For instance, by mimicking the simple movements of worms, researchers have shown the potential of such robots in constrained environments, e.g., movement in narrow cave systems [5,6]. With continued development, autonomous soft robots may be able to perform various complex tasks and execute increasingly complex functions by integrating functional electronic components [7]. In addition, soft robots can achieve desired applications in a controllable manner without coupling additional instruments, e.g., clamping [8,9,10], crawling [11,12,13], and rolling [14,15]. Currently, the driving structure of soft robots can convert external stimuli, e.g., light [16,17,18], electricity [19,20], magnetism [21], pH [22], temperature [23,24,25], and humidity [26,27,28], into mechanical energy, thereby resulting in macroscopic two-dimensional (2D) or three-dimensional (3D) shape changes. Among these external stimuli, light-driven actuation, where light energy can be converted to mechanical energy, is considered the best choice to realize small, unconstrained, and bionic stimuli-responsive soft actuators due to some unique properties. For example, the light actuation method can be controlled wirelessly, instantly, and remotely, and it exhibits advantages in terms of energy, wavelength selectivity, and eco-friendliness. In addition, we can control the shape deformation of the actuator accurately by changing the physical parameters of the stimulus light source (e.g., intensity, wavelength, and polarization) [29,30]. According to the response mechanism, light stimuli-responsive actuators can be divided into two categories, i.e., photochemical and photothermal effects [31]. Based on the expansion or contraction caused by a photothermal effect, various photothermal response actuators have been prepared. Such actuators have the advantages of unique controllability, fast response ability, reversibility, good repeatability, and high stability [32,33]. Thus, the use of fast photothermal response materials for soft robot design has received increasing attention in recent years.

However, in the soft robot context, actuation without functional applications is flawed [34]. The literature on soft robot technology shows that most current soft robot research focuses on only the study of stimulus-response actuators. In other words, most studies have ignored the functional integration of soft actuators, e.g., sensing and lighting functions. Such functional applications require the integration of sensors or other electronic components with unique functions on soft actuators. However, due to the large deformation of the soft structure of the actuator, conventional electronic components previously used on the rigid substrate cannot be integrated into a soft robot easily [35,36]. Thus, the large deformation of the soft actuator under external stimulation will affect the accuracy of sensor measurements and may potentially compromise or damage the device [37,38]. Currently, there are several methods to realize the integration of soft actuators and functional devices [39,40,41,42,43]. For example, a variety of ion conduction and fluid characteristics can be integrated into the soft drive system using embedded 3D printing technology to prepare an effective soft-sensing actuator [44]. In addition, replacing traditional rigid electronic sensors with electronic sensors based on flexible materials can achieve built-in strain-sensing capabilities, e.g., efficient control of soft robots and functional obstacle detection; however, the 3D printing process is complex, and the types of electronic devices that can be integrated are limited due to the inherent characteristics of the 3D printing process. Electronic devices based on flexible polymer materials are currently in development; however, most exhibit poor performance, and the stability and reliability must be improved for effective practical application [45]. Thus, we must develop a soft robot that can integrate electronic devices to achieve specific functions without impacting the motion of typical structural deformation.

In this study, to solve this problem, we fabricate a preparation scheme for a light stimuli-response soft robot. Here, we select traditional polyimide (PI) commonly used in integrated electronic components as the passive layer, PDMS with a large difference in thermal expansion coefficient and PI as the active layer, and CNT as the photothermal conversion material. The three elements together form a CNT-PDMS/PDMS/PI three-layer light stimulus response film, which is the main structure of the robot. In addition, laser cutting is utilized to form micro-damage in the three-layer film to control the film’s direction and amplitude. Then, the effects of the laser cutting parameters (cutting depth, line spacing, cutting line angle of the laser cutting line, and the depth and width of the cutting groove) on the bending deformation behavior of the film are studied systematically. We find that the influence mechanism of the cutting line and the groove on the bending deformation of the film differs, and the final bending action is also different.

Based on the reversible and stable driving behavior of the CNT-PDMS/PDMS/PI three-layer light stimuli-responsive film after laser cutting, we realize various complex actions by designing simple laser cutting patterns and cutting parameters, including 3D shapes, e.g., tendril, U shape, flat arch bridge, heart, droplet, triangle, and quadrilateral. To demonstrate the practical application of the proposed film, we prepared a self-rolling robot that realizes continuous movement to an area outside a spot under vertical and uniform near-infrared light irradiation. In addition, we prepared a bionic worm robot integrated with multiple commercial LEDs. To ensure normal operation of the LEDs, the robot crawls forward continuously under the stimulation of vertical pulsed light. Collectively, our evaluations and practical demonstrations verify that it is feasible to integrate functional devices into soft actuators, which provides a feasible solution for the development of soft robots with complex functions.

## 2. Materials and Methods

### 2.1. Materials

The PDMS (Sylgard 184 silicone elastomer, Dow Corning) material was purchased from Shenzhen Dow Technology Co., Ltd. (Shenzhen, China). Multi-wall carbon nanotubes (SWCNTs) were purchased from Jiangsu Xianfeng Nanomaterial Technology Co., Ltd. (Nanjing, China), with a diameter of 10–20 nm. Isopropyl alcohol (IPA) was provided by Shanghai Maclin Biochemical Technology Co., Ltd., analytical grade (Shanghai, China).

### 2.2. Preparation of CNT-PDMS Composite Films

First, the solvent isopropyl alcohol was weighed and carbon nanotubes accounting for 3% of the mass fraction of the mixed material (the reason for the selection is shown in Appendix A in the Appendix A) were added. After ultrasonic crushing and stirring several cycles, the carbon nanotubes were evenly dispersed in the solvent. The PDMS base liquid was then added to the mixed solution and dispersed uniformly by ultrasound. After that, the temperature of the solution was controlled at 80 °C until the solvent was close to complete volatilization. Next, the PDMS curing agent was added (accounting for 10% of the PDMS matrix), stirred well, and then put into a vacuum drying oven for vacuum defoaming. Finally, the mixture was scraped onto a glass plate and then cured at 80 °C for 2 h to obtain the CNT-PDMS single-layer composite film.

### 2.3. Preparation of CNT-PDMS/PDMS/PI Three-Layer Composite Film

First, the PI film was smoothly fixed on the glass plate coated with a thin layer of PDMS. After that, the fully stirred and vacuum-defoamed PDMS was scraped onto PI and cured at 80 °C for 2 h. Finally, the previously evenly dispersed CNT/PDMS mixture was scraped onto the PDMS and then cured at 80 °C for 2 h. In this way, the CNT-PDMS/PDMS/PI three-layer composite film was obtained. The film was laser-cut for performance testing.

### 2.4. Characterization and Measurement

The cross-section of the film was observed using a Zeiss high-resolution field emission scanning electron microscope to explore the interface bonding ability of the three layers. Raman spectra of CNT-PDMS and pure PDMS films were obtained using a HORIBA Raman microscope with an excitation wavelength of 532 nm. The near-infrared light source used was a semiconductor laser integrated light source with a wavelength of 808 nm provided by Hite Optoelectronics Co., Ltd. (Beijing, China), and its power density was measured by a TP100 optical power meter. The surface temperature distribution and maximum temperature of the composite were measured and recorded by a T650sc infrared imager and paperless recorder. The bending deformation behavior of the film was recorded using an industrial camera, and the bending performance of the film was analyzed using the curvature calculation formula. The thermal expansion coefficient (CTE), thermal conductivity coefficient, and elastic modulus of each single layer film were measured by a thermomechanical analyzer (TMA), thermal conductivity meter, and intelligent electronic tension machine, respectively.

## 3. Results and Discussion

### 3.1. Morphology Characterization and Light Actuation of CNT-PDMS/PDMS/PI Three-Layer Film

Figure 1a illustrates the process used to prepare the CNT-PDMS/PDMS/PI three-layer film. Initially, the commercial PI film was fixed on the substrate, and then the PDMS and CNT-PDMS films were prepared successively on the PI substrates via blade coating. Then, a three-layer film with good adhesion was formed by a subsequent curing process at 80 °C. Finally, laser processing was utilized to form the actuators with different structural characteristics and functions. Here, we employed ultraviolet femtosecond laser cutting for processing because femtosecond laser direct writing technology has fast processing speed, high precision (micro to nano level), and 3D structure processing. In addition, the thermal effect of the femtosecond laser in cutting engineering is low; thus, it can avoid deformations caused by the thermal stress generated in the laser cutting process, which is beneficial to action control. A scanning electron microscopy (SEM) image of the CNT-PDMS/PDMS/PI actuator is shown in Figure 1b. As shown by the cross-sectional view, the actuator comprises two distinct layers, i.e., CNT-PDMS and PDMS layers at the top and the PI film at the bottom. Note that the boundary between the CNT-PDMS and PDMS layers in the SEM image is effectively indistinguishable, and appears as a single layer with a thickness of 200 μm in the SEM image. Here, the thickness of the PI layer is 50 μm. The interface between the PDMS and PI is clear, the binding is very good, and no stratification phenomenon is observed. Laser cutting creates a V-shaped groove on the actuator. Here, the width and depth of this groove can be controlled by adjusting the laser parameters. For example, the maximum width of the groove formed by laser cutting along a single line is only 50 μm (shown in Figure 1b). The depth of the V-shaped groove and the distance between two adjacent grooves are defined as the cutting depth and line spacing of the laser cutting, respectively. Figure 1c shows the Raman spectra of the CNT-PDMS film and pure PDMS film excited at 532 nm. Compared with the Raman absorption spectra of PDMS, the Raman absorption peaks of carbon nanotubes, D mode (defect mode) near 1347 cm^−1^, G mode (tangential tensile mode) near 1584 cm^−1^, and 2D mode near 2701 cm^−1^ all appear in the Raman absorption spectra of CNT-PDMS composite films, which proves that carbon nanotubes are dispersed uniformly in the PDMS without chemical changes [46,47].

Figure 1d shows the bending process of the actuator with time under near-infrared (NIR) light. When the light is irradiated, the temperature of the actuator increases rapidly according to the ultra-high photothermal conversion effect of CNT. Note that the thermal expansion coefficient of pure PDMS is 370 ppm/K (Appendix A), and the thermal expansion coefficient of the CNT-PDMS layer doped with CNT is reduced to 307 ppm/K; however, it remains much greater than that of PI (33 ppm/K). Thus, the actuator will bend to the PI side. When the NIR light irradiates the sample, the temperature increases rapidly, and then the actuator bends accordingly, which also shows a rapid increase stage. After turning off the light, the temperature drops rapidly. According to the heat transfer rate, the temperature drop trend is first fast and then slow, gradually approaching room temperature. In addition, the actuator returns to the initial shape when the temperature returns to room temperature, which means that the light actuation of the actuator has a fast response and good reversibility. To verify the cycle performance of the actuator, the repeatability of the CNT-PDMS/PDMS/PI actuator was further tested under NIR light of 400 mW/cm^2^. The experimental results are shown in Figure 1e. As shown, after 50 bending/nonbending cycles, the bending curvature under illumination and the initial curvature under nonillumination do not change, which proves that the actuator has good reversible repeatability for NIR light.

### 3.2. Effect of Laser Cutting on Bending Performance of CNT-PDMS/PDMS/PI Actuator

Here, we investigate the effects of laser cutting parameters (e.g., cutting depth and line spacing of the cutting line, and the depth and width of the cutting groove) on the bending behavior of the CNT-PDMS/PDMS/PI actuator under NIR light illumination. Figure 2 shows the effect of the laser cutting line on the bending performance of the CNT-PDMS/PDMS/PI actuator under NIR irradiation with a power density of 400 mW/cm^2^ (unless noted, the power density of the near-infrared light source used in this paper is 400 mW/cm^2^). The effect of laser cutting depth on the bending performance of the actuator under constant line spacing of the laser cutting line is shown in Figure 2. According to Timoshenko’s beam theory (shown in Appendix A in the Appendix A) [48], which is always used to explain the photothermal expansion mechanism, the deformation curvature of the film is only related to the temperature difference and elastic modulus on the basis of the fixed material and film size. The bending curvature of the original uncut film under NIR light irradiation is 0.085 mm^−1^ (the bending curvature calculation method is shown in Appendix A and Appendix A in the Appendix A). As the cutting depth increases, the bending curvature of the actuator initially increases and then decreases, and when the cutting depth is 205 μm (i.e., the total thickness of the CNT-PDMS and PDMS layers), the curvature reaches a maximum of 0.13 mm^−1^, which is approximately 1.5 times that of the original film. When the cutting line depth does not reach the PI film, the bending curvature of the film increases as the cutting line depth increases because the generation of microgrooves reduces the bending stiffness of the film. In addition, with increased cutting depth, the bending stiffness decreases gradually, and a greater bending curvature can be obtained. At this time, the bending deformation behavior of the film is that, under light, it will curl along the long axis of the film to form a short cylinder (as shown in the upper left corner of Figure 2a). When the depth of the cutting line reaches the PI film, the curvature drops rapidly to close to 0. In other words, no curling occurs (as shown in the lower right corner of Figure 2a). Curling does not occur because the area between the two adjacent cutting lines becomes an independent area when the depth of the cutting line reaches the PI film, and the adjacent cutting lines will curl along the long axis in the previous manner. Locally, there is a tendency to curl along the vertical direction. However, the overall bending stiffness in this direction is large, and the final effect is that no curling phenomenon is observed. Figure 2b shows the effect of cutting line spacing on the bending performance of the actuator while controlling the cutting depth. As can be seen, with increasing cutting line spacing, the bending curvature of the film under illumination continues to decrease, and the overall trend is linear because the V-shaped micro-groove caused by a single cutting line can reduce the overall bending stiffness of the film, and an increasing number of grooves causes additional reduction.

Based on the above observations, we then explore the effect of laser cutting of the passive layer on the bending performance of the CNT-PDMS/PDMS/PI actuator. As shown in Figure 2c, we design concentrated patterns with high stretchability, i.e., a negative Poisson’s ratio structure showing expansion under tensile load, an origami structure, horizontal stripes, and vertical stripes. As shown in Figure 2d, the laser cutting of the passive layer can increase the bending curvature of the film, and obvious bending deformation can even be achieved under sunlight conditions (the power density of standard sunlight [49] is 100 mW/cm^2^). We speculate that the passive layer laser cutting, similar to the active layer laser cutting, can reduce the overall bending stiffness of the film such that the bending deformation of the actuator is improved. However, different patterns may have different degrees of reduction; thus, the magnitude of the deformation also differed. In contrast, the overall cutting along the horizontal axis of the film reduces the bending stiffness most, which makes the bending curvature of the film reach a maximum value of 0.050 mm^−1^ under sunlight irradiation.

Then, we laser cut the three-layer film at different angles (0°, 15°, 30°, 45°, 60°, 75°, and 90°) along the horizontal axis. This designed structure realizes the controllability of the actuator in the bending direction. As can be seen from the optical images in Figure 2e, which show the bending deformation of the actuator with different cutting line angles under 808 nm NIR illumination, the curling direction of the actuator is perpendicular to the direction of the cutting line. When the angle of the cutting line is 0°, i.e., when cutting horizontally, the actuator curls along the long axis to form a short cylinder. In addition, at a cutting line angle of 90°, i.e., longitudinal cutting, the actuator curls along the long axis direction to form a short cylinder, and when there is an angle between the cutting line and the film spindle, the actuator can curl into a spiral shape. With an increasing cutting line angle, the diameter of the spiral decreases, and the pitch becomes larger. In addition, we establish a finite element model (shown in Appendix A and Appendix A in the Appendix A). In the simulation, we use a fixed temperature field instead of light irradiation to study the theoretical bending deformation shape of the film, and the calculation results are shown in Figure 2e. It is found that the effect of the cutting line angle on the bending shape of the actuator in the simulation results is basically consistent with the experimental results (the direction of the cutting line and the direction of the spindle will not necessarily cause the distortion of the actuator). This proves that the finite element model can predict the actuator’s action accurately.

### 3.3. Effect of Groove on Bending Performance of CNT-PDMS/PDMS/PI Actuator

When multiple laser cutting lines are superimposed to form a nonnegligible groove, the influence of this groove on the actuator’s bending performance is not known. Thus, here, we discuss the effect of grooves on the bending performance of the CNT-PDMS/PDMS/PI actuator.

Figure 3a shows the effect of groove depth on the actuator’s bending performance with a groove width fixed at 0.45 mm. As can be seen, as the groove depth increases, the bending curvature of the actuator initially increases and then decreases. When the cutting depth reaches 101 μm (i.e., the thickness of the CNT-PDMS layer), the bending curvature of the actuator reaches the maximum value of 0.090 mm^−1^ because, for the actuator with grooves, the bending performance is primarily affected by the temperature increase and bending stiffness. When the groove depth is less than the thickness of the CNT-PDMS layer, i.e., the CNT-PDMS layer has not been cut through, the temperature increase in the actuator during illumination does not change significantly; however, the bending stiffness is reduced significantly. Thus, the curvature of the actuator demonstrates an upward trend. When the groove depth is greater than the thickness of the CNT-PDMS layer, there is no carbon material in the groove area where the temperature increase is small under illumination. In addition, because there is no carbon material in this region, the thermal conductivity will be reduced, which affects the heat conduction in different regions. This is also verified in experiments and finite element simulations (Figure 3c–f). Therefore, the bending curvature of the film decreases.

Figure 3b shows the effect of groove width on the actuator’s bending performance with controlled groove depths. As shown, the bending curvature change in the actuator exhibits two distinct mechanisms due to the different groove depths. At a groove depth of 60 μm, i.e., the laser cutting does not penetrate the PDMS layer, the change in the bending curvature with the groove width is not obvious. In contrast, when the laser cutting damages the PDMS layer, the bending curvature of the actuator is reduced as the groove width increases because, as the groove width increases, the actual photothermal conversion layer area is reduced, which leads to a reduction in the corresponding temperature increase. Thus, the actuator’s bending curvature is reduced continuously.

In addition, the presence of the groove gives the actuator a new function: thermal insulation function. We used an IR thermal imaging camera to record the IR image of the actuator’s temperature change without grooves and with a 0.5 mm wide groove under NIR illumination (shown in Figure 3c). As shown, for the actuator without a groove, the temperature distribution is symmetrical relative to the spot center. In contrast, for the actuator with a groove, the temperature of the groove area exhibits a cliff-like reduction compared to the symmetrical area, and an obvious temperature boundary can be observed. To better understand the groove insulation mechanism, a typical thermal simulation numerical method, finite element model, is established using the COMSOL software (COMSOL Multiphysics 5.6), and the temperature field of the film is calculated based on the thermal conductivity of each single layer film (shown in Appendix A in the Appendix A). From the simulation results shown in Figure 3d, when there is no groove, the maximum and minimum temperatures of the actuator are 112 °C and 37 °C, respectively, and the temperature decreases gradually from the center to the boundary. When the groove is present, the maximum temperature and minimum temperature of the actuator are 112 °C and 25 °C, respectively, and the temperature drops in a gradient at the groove. Figure 3e,f show the curves of temperature and distance from the heat source with and without grooves. As can be seen from the curve of Figure 3e, when the distance from the heat source is 0.7–1.2 mm (i.e., the location of the groove), the slope of the curve is much steeper than that without the groove. In addition, when the distance from the heat source is 1.2 mm, the temperature with the groove is 27 °C, and the temperature without the groove is 43.65 °C. These findings prove that the groove exhibits effective thermal insulation. The thermal insulation function of the groove also explains that the existence of grooves reduces the overall temperature of the film and thus reduces the bending curvature.

## 4. Practical Applications of CNT-PDMS/PDMS/PI Actuator

By controlling the processing mode, the previously mentioned light-driven strategy can be applied to shape-programmable actuators with complex shape-adjustable deformations. Benefiting from the fast and high-precision laser cutting and the programmable design of the laser patterns, multiple complex patterns can be designed to realize various complex motion modes to expand the range of 3D reconfigurable structures. Figure 4a summarizes the experimental results of several simple representative 2D pattern designs that can be converted into 3D shapes. In these 2D CAD models, the different colors represent different cutting depths. Here, black, blue, and green represent slight laser cutting with increasing depths of 50 μm, 100 μm, and 150 μm, respectively. The red part indicates that the laser cutting damaged the PI layer (240 μm). Note that all structures were prepared using a CNT-PDMS active layer (100 μm, 3 wt% CNT), a PDMS intermediate layer (100 μm), and a PI passive layer (50 μm). In addition, the light-triggered deformation is performed at a power density of 1000 mW/cm^2^ and a wavelength of 808 nm. By adjusting the laser cutting parameters, we obtained various three-dimensional special shapes, such as tendril, U shape, flat arch bridge, heart, droplet, triangle, and quadrilateral (shown in Figure 4a). Here, we find that increasingly complex actions can be realized by optimizing more diverse combinations of cutting parameters.

In addition to these static illumination shapes, we also developed actuators that can roll continuously under vertical NIR light, as shown in Figure 4b,c. Here, we connected five segments of the same size film with different cutting depths and line spacings through grooves, and we connected the two ends of the film to form a ring using a physical method. The corresponding laser-cutting diagram is shown in the illustration in the upper left corner of Figure 4b. Here, the PI membrane faces the light source. At room temperature, the ring is only subjected to gravity and is initially symmetrical. Under NIR light irradiation, the interfacial stress drives the curvature of the film irradiated by light to become smaller, and the viewing angle effect tends to be flat. However, due to the different cutting depths and line spacings of the three parts, the bending amplitude during illumination differs, i.e., the degree of flatness differs. As shown in Figure 4b, under illumination, the grooves in each region are subjected to interfacial stress (F1, F2, F2′, and F3). However, since the cutting depth of the three areas receiving light decreases from left to right, and the cutting line spacing increases, the interface stress gradually decreases from left to right, i.e., F1 > F2 = F2′ > F3. After force synthesis, the additional force borne by the actuator is the direction of the force to the right. Thus, the ring tilts to the right, and the center of gravity shifts, which results in a large torque to the right. In turn, this forces the film to move to the side with low cutting depth and large line spacing. When the illumination part moves to the lower end, the actuator returns to the original curved shape, thereby providing tension to further promote the ring’s movement. Thus, under continuous vertical light illumination, the light-driven rolling is continuous. As shown in Figure 4c and Appendix A, when the light is on, the ring can roll forward 10 mm in 25 s. Until part of the ring leaves the illumination area, the interface stress on the actuator basically disappears, and the actuator stops rolling.

We also designed a crawling actuator that imitates worms. Note that walking is the most common mode of movement in biological systems, where the driving force for progress comes from periodic body deformation and asymmetric friction. For light stimuli-responsive actuators, periodic body deformation can be realized by applying simple periodic external stimuli, and asymmetric friction can be achieved using tilted light. However, when the light is irradiated vertically (shown in Figure 5a and Appendix A), the traditional film is curved uniformly due to the uniform distribution of the film, and the friction on the left and right ends is also consistent. When the light is turned off, the actuator will return to the initial position and cannot achieve forward crawling. Thus, asymmetric deformation is a necessary condition to achieve crawling. Here, we can realize asymmetric friction via laser cutting, where the left half is damaged by laser cutting up to the PI layer, and the right half is cut slightly. When the light is irradiated vertically, the right half of the actuator bends and the left half does not change. Thus, the angle of the right half part contacting the ground is closer to vertical and is relatively flat on the ground. In contrast, the tilt angle of the left half part is larger than the right half part, which reduces the surface contact of the left half part relative to the right half part. Thus, the friction on the right half part is greater than that on the left half part (F1 > F2), and the left half part moves forward. When the light stops irradiating, the actuator returns to its original flat shape. At this point, the friction on the right half part is reduced; however, the left half part exhibits no change (F2′ > F1′), which causes the right half part to move forward. By turning the NIR light on and off in turn, the crawling actuator can be driven to move forward (shown in Figure 5b and Appendix A).

In addition, we can control the speed of the actuator crawling precisely by controlling the area and depth of the laser cutting. As shown in Figure 5c, when we change the ratio of the length of the left and right parts (i.e., the black and red parts) while controlling the cutting depth, the crawling distance increases initially and then decreases. When the length ratio is 1:4, the crawling distance of the actuator is 0 because the actuating part (i.e., the right half) of the design is too short, which results in an insufficient driving force. When the NIR light is irradiated, the right half cannot pull the left half. When the length ratio is 1:2, the actuator begins to crawl forward, and the crawling distance becomes greater as the length ratio increases. At a length ratio of 2:1, the crawling distance reaches the maximum value of 4.5 mm; however, when the length ratio continues to increase to 4:1, the crawling distance is reduced to only 1.5 mm because, in this case, the left half of the actuator, i.e., fixed area, is too short, which results in insufficient friction on the left end when the NIR light is turned off. Here, the left leg does move backward while the right leg is moving forward, which reduces the actuator’s forward crawling distance. Figure 5d shows the effect of the cutting depth of the right half on the actuator’s crawling speed. As can be seen, the distance of the actuator’s forward movement in a switching optical cycle increases as the cutting depth increases.

To verify the ability of the worm-like robot to integrate or load functional electronic devices, we integrated and lit four commercial LEDs in series in the passive layer (i.e., the PI layer), as shown in Figure 5e and Appendix A. Here, we integrate the LEDs in series in the left half of the crawler actuator. When the current is on, the LEDs emit different colors of light (shown in Figure 5e). In addition, while the actuator is crawling forward, the LEDs remain illuminated and are not damaged because, during the actuator’s movement, the left half of the integrated LEDs do not bend and deform due to laser cutting damage to the PI layer. Thus, during the crawling process, there is no obvious deformation at the integrated LEDs, which avoids damage caused by the large strain of the LEDs. The LED devices we selected are commercial electronic devices with large size and weight. Even under heavy loads, the soft robot can still achieve the crawling of the actuator and the normal operation of the electronic devices (light-emitting function). It can be inferred that we can integrate other functional devices to achieve various functions to meet the growing need for controllable operation and complex functional applications in complex scenes, which represents significant progress in soft robot research. The actuator can perform single or multi-modal functions during dynamic deformation, e.g., sensing, luminescence, and communication.

## 5. Conclusions

In this paper, we demonstrated a simple and generalizable method to fabricate NIR light-driven soft robots. Here, the complex shape of the actuator is controlled by high-precision ultraviolet femtosecond laser processing. Using this method, we developed several unique 3D deformation structures with different operating modes, e.g., rolling and crawling. In addition, we presented a practical application of a bionic worm integrated with multiple LEDs to verify the feasibility of integrating functional electronic devices on the software robot developed in this study. This is a general solution for the integration of electrical functions for soft robots. Various electronic components, even integrated circuits, can be integrated into flexible robots, which provides a practical solution to enrich the functions of soft robots and expand the application fields. The integration of actuation and function effectively improves the agility, adaptability, and universality of soft robots in various interdisciplinary fields, e.g., intelligent manufacturing, medical surgery, and bionics.

## Figures and Tables

**Figure 1 micromachines-15-00534-f001:**
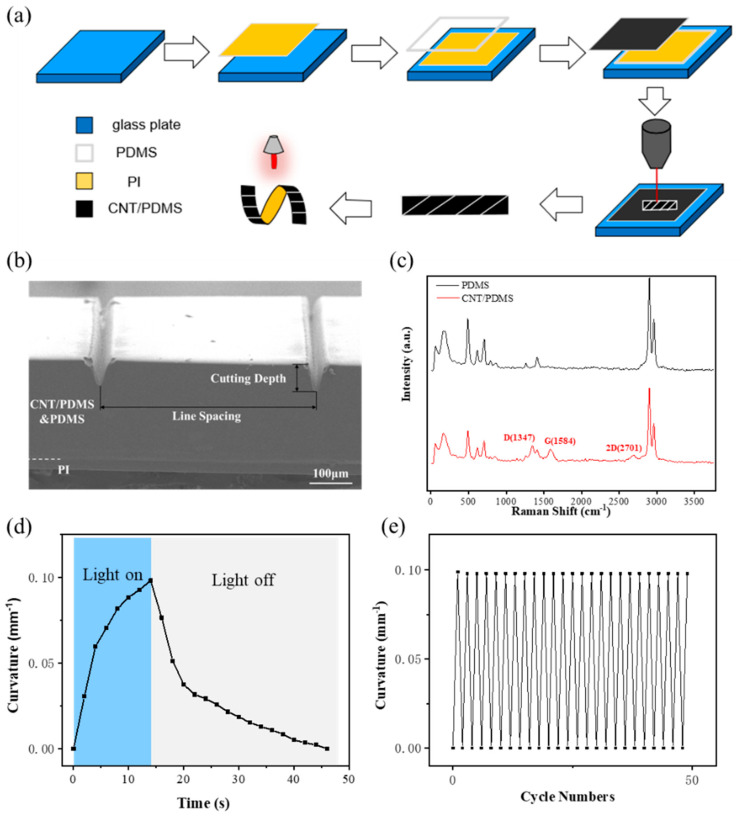
Preparation process and light actuation performance of CNT− PDMS/PDMS/PI actuator. (**a**) Schematic illustrations of the process flow to fabricate the controllable CNT−PDMS/PDMS/PI actuator. (**b**) SEM image (cross−sectional view) of the CNT−PDMS/PDMS/PI actuator. (**c**) Raman spectroscopy of CNT−PDMS and PDMS layers. (**d**) Time-bending curve of CNT−PDMS/PDMS/PI actuator before and after NIR light illumination. (**e**) Repeatability test of the photothermal responsive CNT−PDMS/PDMS/PI actuator toward 808 nm NIR light irradiation.

**Figure 2 micromachines-15-00534-f002:**
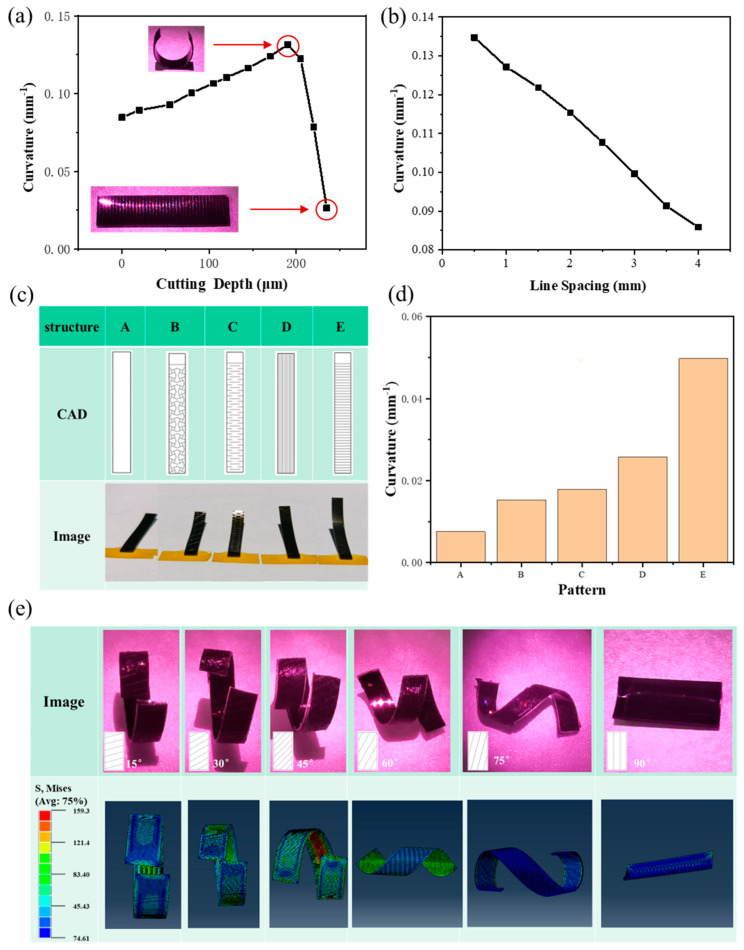
Effect of laser cutting line on bending performance of CNT−PDMS/PDMS/PI actuator. (**a**) Actuation performance of actuators with different cutting depths toward 808 nm NIR illumination. (**b**) Actuation performance of actuators with different line spacing toward 808 nm NIR illumination. (**c**) CAD drawing and optical images of actuators with different passive layer cutting patterns (structure A is an uncut structure, B is a negative Poisson’s ratio structure, C is an origami structure, D is a vertical stripe, and E is a horizontal stripe). (**d**) Actuation performance of actuators with different passive layer cutting patterns toward sunlight illumination. (**e**) Optical images and finite element simulations of actuators with cutting lines at different angles toward 808 nm NIR illumination.

**Figure 3 micromachines-15-00534-f003:**
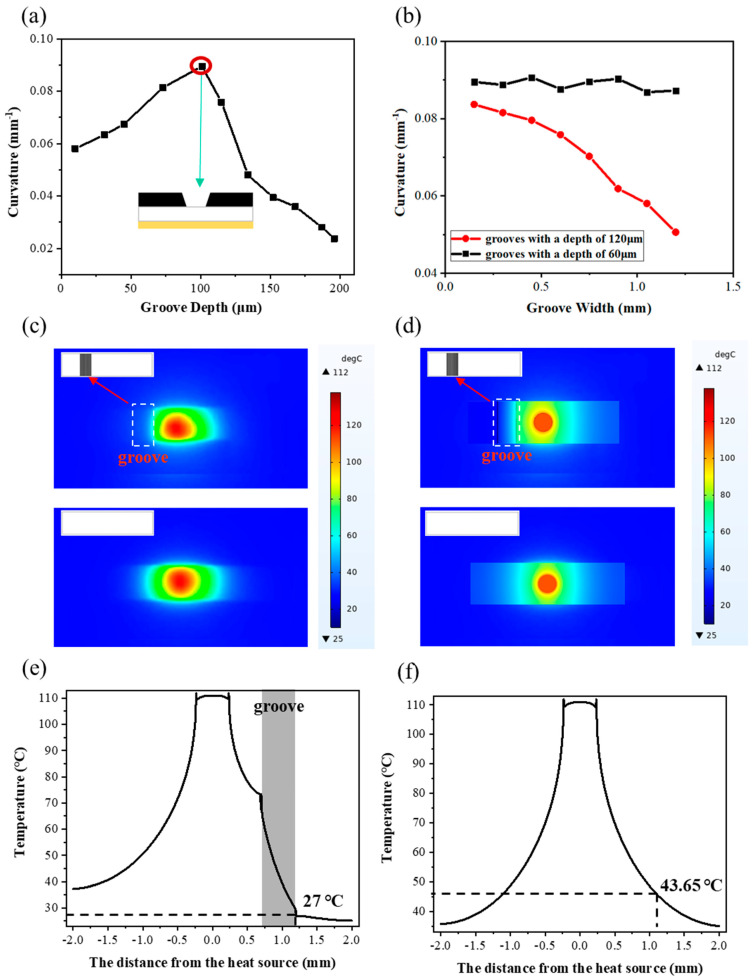
Effect of groove on bending performance of CNT−PDMS/PDMS/PI actuator. (**a**) Actuation performance of actuators with different groove depths toward 808 nm NIR light illumination. (**b**) Actuation performance of actuators with different groove widths toward 808 nm NIR light illumination. (**c**) IR photographs and (**d**) finite element simulations of actuators with and without grooves toward 808 nm vertical NIR illumination (the top left illustration is a CAD drawing of the actuator with and without grooves; the film size is 1 × 4 mm, the groove width is 0.5 mm, the spot center is in the middle of the film, the groove is 0.7 mm away from the spot center, and the white border marks the location of the groove). Temperature profiles of membranes (**e**) with and (**f**) without grooves.

**Figure 4 micromachines-15-00534-f004:**
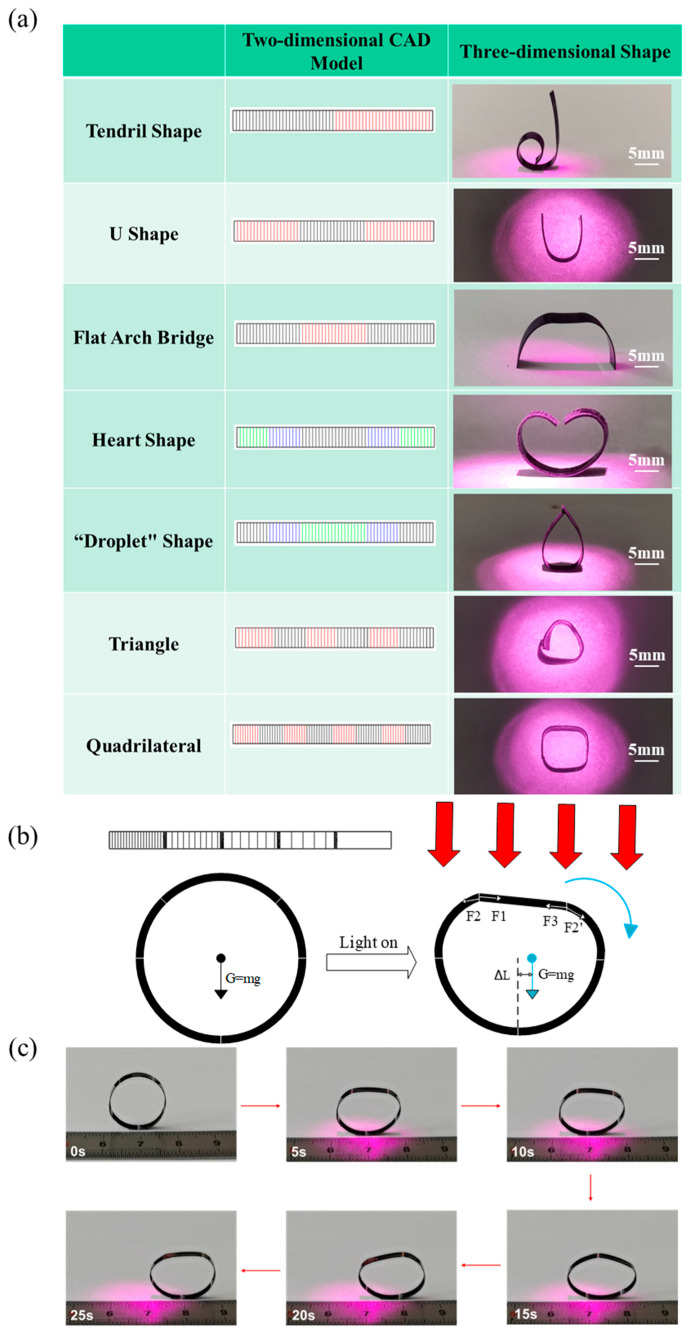
Practical applications of CNT-PDMS/PDMS/PI actuator. (**a**) 2D CAD models and optical images of actuators with different patterns toward vertical NIR light illumination. Here, black, blue, and green represent slight laser cutting with increasing depths (50 μm, 100 μm, and 150 μm, respectively). The red region indicates that laser cutting damaged the PI layer (240 μm). (**b**) Force analysis of soft rolling robot in different positions (F1, F2, F2′, and F3 are interfacial stress; ΔL is the distance that the actuator moves; Arrows represent the direction of the force). (**c**) Photographs showing light-driven rolling of actuators with different grooves and cutting lines toward vertical NIR light illumination.

**Figure 5 micromachines-15-00534-f005:**
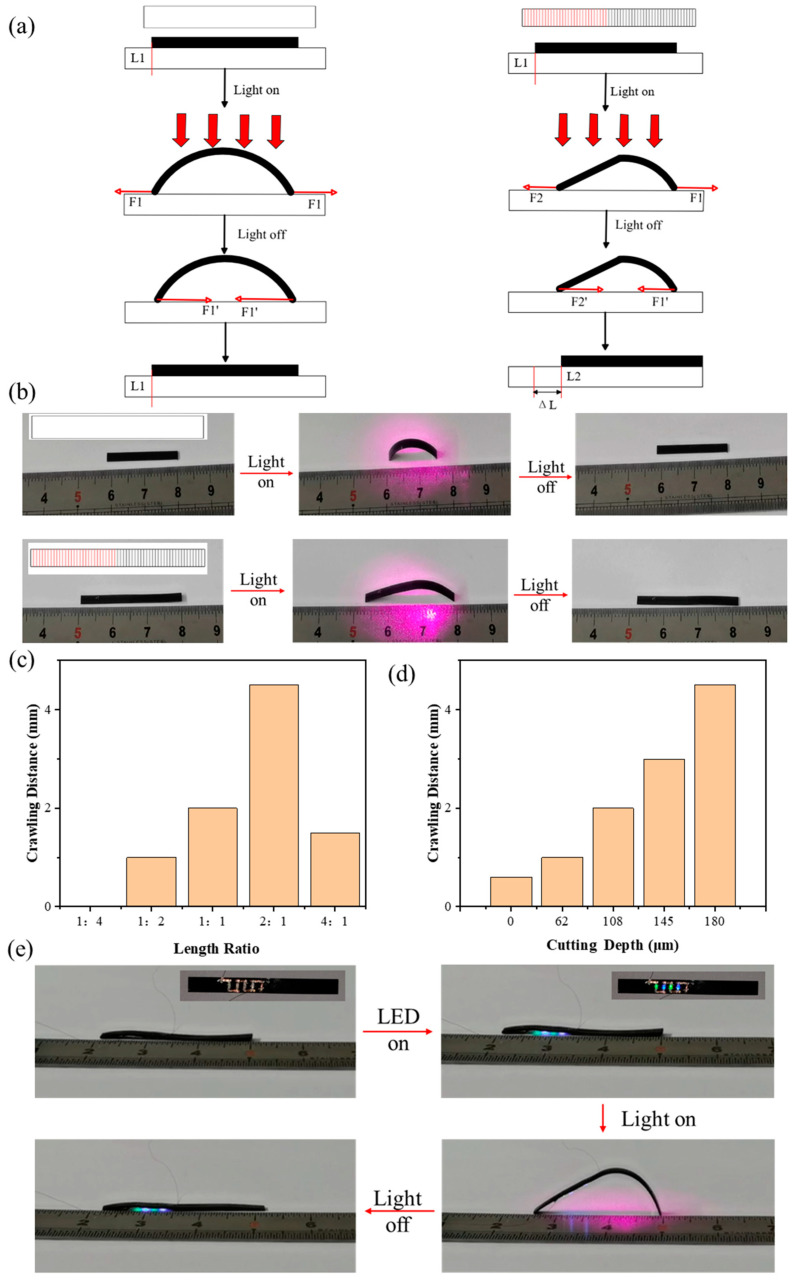
Crawling actuation of CNT-PDMS/PDMS/PI actuator. (**a**) Force analysis of soft crawling robot in different positions. Here, red represents the cutting depth to the passive layer, and black indicates that the cutting depth does not reach the passive layer (Arrows represent the direction of the force). (**b**) Photographs of light-driven crawling of actuators with and without cutting lines in a process of switching vertical NIR light. (**c**) Crawling distance curve of actuators with different length ratios of black area to red area in a process of switching vertical NIR light. (**d**) Crawling distance curve of actuators with different cutting depths in the black area in a process of switching vertical NIR light. (**e**) Crawling actuation of the actuator loaded with LEDs in a process of switching vertical light irradiation.

## Data Availability

All data required to evaluate the conclusions of this study are provided in the article and the Appendix A.

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
