# Peer review of "Light-Responsive Soft Robot Integrating Actuation and Function Based on Laser Cutting"

_micromachines, 2024, doi:10.3390/mi15040534_

Round 1

Reviewer 1 Report

Comments and Suggestions for Authors

This manuscript reports on the development of light-responsive soft actuators prepared by laser cutting. By adjusting the cutting depth, cutting width, and the spacing of the cutting line, the actuators can exhibit different bending curvatures and thus be used to fabricate worm-like soft robots. Although the authors have studied these aspects in detail, the novelty of this work needs to be reinforced before it can be considered for publication.

According to the authors, current research on soft robots has been largely limited to the study of actuator deformation and motion at the expense of practical applications (which the authors claim is the novelty of this manuscript). However, the soft robots developed in this manuscript also do not demonstrate specific practical applications, again focusing primarily on the study of soft robot deformation and motion. In Figure 5e, they only show a worm-like soft robot integrated with LEDs, which is not enough to demonstrate practical applications. In order to strengthen the novelty of this manuscript, practical application scenarios of the developed soft robot should be further explored.

Comments on the Quality of English Language

The author needs to make minor changes to the English language.

Reviewer 2 Report

Comments and Suggestions for Authors

The author provides a well-organized introduction. Varying the mechanical design of grooves using the laser cutting process utilizes efficient fabrication and precise comparison. 

The author argues that actuation without functional application is flawed. However, the research also provides only deformability with varying designs and crawling, which is a general demonstration of soft robotics.

How was the 3% of CNT-PDMS chosen for the experiment?

Remove non-scientific words like 'certain amount' and 'several cycles'.

Can't understand the 'PDMS substrate' in 2.2.

Author should kindly provide details of simulation in Fig.2(d). 

Check materials in the paper. For example, there is no figure 2(e). 

Enlarge words in Figures. All words and features must be visible and understandable.

In Figure 3(b), the x-axis title is 'groove width'. However, the plot shows images with varying depths.

Add details of the images and white boxes in Figure 3(c)(d). Remove worthless space of the Figure 3(c)(d)

The author must carefully adjust the images in all figures with balance. The author should make the picture larger, as the information is more important.
